# Knowledge, attitude, and hesitancy towards COVID-19 vaccine among university students of Bangladesh

Md. Mostafizur Rahman[1]*, Musabber Ali Chisty[2], Md. Ashraful Alam[3], Mohammed Sadman Sakib[1‡], Masrur Abdul Quader[1‡], Ifta Alam Shobuj[1‡], Md. Abdul Halim[1‡], Farzana Rahman[4‡]

1 Department of Disaster and Human Security Management, Faculty of Arts and Social Sciences, Bangladesh University of Professionals, Mirpur Cantonment, Dhaka, Bangladesh, 2 Institute of Disaster Management and Vulnerability Studies, University of Dhaka, Dhaka, Bangladesh, 3 Department of Global Health Policy, Graduate School of Medicine, The University of Tokyo, Tokyo, Japan, 4 Department of Computer Science and Engineering, Independent University, Bangladesh, Dhaka, Bangladesh

☯ These authors contributed equally to this work.
‡ MSS, MAQ, IAS, MAH and FR also contributed equally to this work.
* mostafizur@bup.edu.com

Data Availability Statement: Data cannot be shared publicly for ethical issue. Data are available for researchers for research purpose only. The researchers can request data to the Ethical Review Committee, Institute of Disaster Management and

## Abstract

Global vaccination coverage is an urgent need to recover the recent pandemic COVID-19. However, people are concerned about the safety and efficacy of this vaccination program. Thus, it has become crucial to examine the knowledge, attitude, and hesitancy towards the vaccine. An online cross-sectional survey was conducted among university students of Bangladesh. Total of 449 university students participated. Most of these students used the internet (34.74%), social media (33.41%), and electronic media (25.61%) as a source of COVID-19 vaccine information. Overall, 58.13% and 64.81% of university students reported positive knowledge and attitude towards the COVID-19 vaccine. 54.34% of these students agreed that the COVID-19 vaccine is safe and effective. 43.88% believed that the vaccine could stop the pandemic. The Spearman's Rank correlation determined the positive correlation between knowledge and attitude. The negative correlation was determined between positive knowledge and hesitancy, and positive attitude and hesitancy. University students with positive knowledge and attitude showed lower hesitancy. Multiple logistic regression analyses determined the university type and degree major as the predictors of knowledge, whereas only degree major was the predictor of attitudes. 26.06% of the study population showed their hesitancy towards the vaccine. University type and degree major were also determined as predictors of this hesitancy. They rated fear of side effects (87.18%) and lack of information (70.94%) as the most reasons for the hesitancy. The findings from this study can aid the ongoing and future COVID-19 vaccination plan for university students. The national and international authorities can have substantial information for a successful inoculation campaign.

Vulnerability Studies, University of Dhaka, Dhaka, Bangladesh at this email address idmvs@du.ac.bd.

**Funding:** The authors received no specific funding for this work.

**Competing interests:** The authors have declared that no competing interests exist.

## Introduction

The World Health Organization declared COVID-19 as a pandemic on March 11, 2020 [1]. Over 200 countries have already been affected by the pandemic [2]. This highly contagious disease has become a burden for the world. Millions of COVID-19 cases and deaths [3] have made the situation uncertain. Global communities continue to battle the public health crises arising from the long-term pandemic. It has also triggered formidable socio-economic and psychological impacts [4]. A safe and effective COVID-19 vaccine has become urgent for every region. Many countries have already started the inoculation campaign for their people [5]. Millions of COVID-19 vaccines have already been administered to stop the pandemic [3]. Several vaccines have already been available to rollout globally [6]. However, a successful inoculation campaign requires enough knowledge and positive attitudes towards the vaccine. A lack of knowledge and attitude may create vaccine hesitancy, defined as "the decision to delay vaccination or the refusal to vaccinate despite available vaccination services" [7]. In the past, vaccine hesitancy has been observed against influenza, human papillomavirus, and pneumococcal vaccines [8–11]. The hesitancy was also found for COVID-19 vaccines in some countries [12–15]. Numerous studies conducted in the Czech Republic, Italy, Lebanon, and Bangladesh revealed that a variety of socio-demographic factors, behavioral patterns, vaccine availability, side effects, a lack of information, or a lack of trust regarding the COVID-19 vaccine might all contribute to the hesitancy [13, 16–19]. COVID-19 vaccine rejection was found to be much higher among the elderly, rural, semi-urban, and slum groups, farmers, day laborers, and homemakers, as well as those with low education and lack of confidence in Bangladesh's healthcare system [20].

The skyrocketing COVID-19 cases in India and Nepal [3] has placed the situation worst in the South-Asian region. As a neighboring country, Bangladesh has also struggled to control the pandemic by applying the lockdown process [21]. Bangladesh's government began providing Covidshield, an Oxford AstraZeneca vaccine, to the general population, and additional vaccines including Moderna, Pfizer, and SinoPharma have also been launched [19, 22]. As of December 8, 2021, 7 vaccines were approved for use in Bangladesh [23]. However, as is the case in many other countries, the government initially focused vaccination efforts on relatively limited categories of people (frontline medical personnel, government employees, private sector officials working on pandemic issues, and people aged 40 years and over), with the anticipation that the entire population would be eligible for vaccinations later [19, 24]. On February 7, 2021, the authority launched a mass COVID-19 vaccination campaign [25]. 106,575,146 vaccines have already been administered, with 24.32% (39,653,764) of the entire population completely vaccinated as of December 8, 2021 [26]. Initially, the authority did not include university students. However, all the universities were closed from the public holidays [27] initiated to control the pandemic. These educational institutes have suffered a lot due to the pandemic [28]. Several strategies, such as online classes and planning to open the institutes maintaining the health behavior, have already been taken to continue the higher education. However, all these strategies faced severe challenges and, in some cases, failed [29]. Therefore, the authority has decided to open the university after an inoculation campaign for university students [30]. However, people are concerned about the safety and efficacy of this vaccination program [31]. Hence, it should be noteworthy that the growing concerns over the vaccine's safety and efficacy may delay this vaccination process. Universities in Bangladesh have reopened in October, 2021 [32]. Nonetheless, all of these students must be vaccinated against COVID-19. A study is necessary to determine these students' knowledge, attitude, and hesitancy (KAH) concerning COVID-19 vaccinations. Apart from the significance of this study, there was a dearth of research on university students, with the exception of a few studies on the general population of Bangladesh [18–20].

The main objective of this study is to evaluate the KAH level towards the COVID-19 vaccine among university students of Bangladesh. The research questions for this study are "What are the knowledge and attitude levels regarding COVID-19 vaccine among these students?", "What are the socio-demographic and academic factors associated with the KAH level of university students towards the COVID-19 vaccine?", and "Which factors could lead these students towards hesitancy regarding the COVID-19 vaccination?".

## Materials and methods

### Research design

The current survey was performed among university students of Bangladesh. They were also divided into five categories based on their majors: Arts and Social Science (an interdisciplinary field of study that encompasses behavioral and social science), Business and Economics (which focuses on business analysis, microeconomics, macroeconomics, financial and management information, and so on), Science and Engineering (which includes students who study diverse subjects such as biological science, physics, chemistry, and other engineering subjects), and Science and Engineering (which includes students who study diverse subjects such as natural science, physics, chemistry, etc.), and Medical Studies.

This cross-sectional study examined the KAH level towards the newly administered COVID-19 vaccine. Due to the ongoing pandemic, a rapid online self-reported KAH survey was carried out. The boundary conditions were university students living in Bangladesh and required internet access. We followed The Checklist for Reporting Results of Internet Surveys (CHERRIES) guidelines [33]. This study was approved by the Ethical Review Committee of the Institute of Disaster Management and Vulnerability Studies, University of Dhaka, Bangladesh (ERC-01/020221). It has maintained all the ethical issues. Consents over the phone and through online conversations were taken from the participants. The cover page of the questionnaire also clearly described the length of time of the survey, confidentiality of the data, the purpose of the study, the research team, and associated ethical issues. There was no incentive for the participants. Respondents were entirely free to leave the online survey at any point.

### Survey tools

Literature reviews regarding the COVID-19 vaccine [24, 34–36] were performed. A draft electronic questionnaire, both in English and translated local Bengali version, was then prepared in Google form. A pilot survey conducted among some university students, experts' opinions, and cultural appropriateness were considered for preparing the final electronic questionnaire. The final questionnaire had five sections with mandatory response items, such as socio-demographic and academic information of the university students; most used sources for COVID-19 vaccine-related information and if they tested COVID-19 positive before; knowledge section; attitude section; and hesitancy section. The hesitancy section was only for the respondents who reported their hesitancy. Total 20 items were in the KAH section (knowledge = 09, attitude = 05 and hesitancy = 06 items). A same scale range of 0–1 was designed for all the items (positive response and hesitancy = 1, neutral response = 0.5 and negative response/don't know/disagree/no hesitancy = 0). The knowledge section had the statements regarding the vaccine's effectiveness, health behavior after 1st dose of the vaccine, side effects, and misconceptions about the vaccine (Table 3). The attitude section contained the pandemic concerns, vaccine safety and effectiveness, and willingness to pursue the vaccine (Table 4). The hesitancy section contained the reasons for hesitancy towards the vaccine. This section addressed the perception of ineligibility to pursue the vaccine, lack of enough information, fear of side effects, the probable cost of the vaccine, religious views, and other reasons (Table 8). We also

calculated the Cronbach's Alpha as 0.86, 0.83, and around 0.60 for the knowledge, attitude, and hesitancy section, respectively. 0.60–0.70 are generally accepted values for internal consistency, where greater than 0.80 determines the excellent level [37, 38].

## Data collection and data analysis

A group of university students was recruited (based on their research experience) to administer the questionnaire from March 12 to April 02, 2021. Several online platforms, namely Facebook, email, WhatsApp, Imo, and Google classroom, were used. We followed the convenience sampling technique. Convenience sampling is a technique for non-probabilistic sampling that is frequently used in clinical research. Typically, this sampling strategy draws clinical cases or participants from an easily accessible area, such as a medical records database, an internet site, or a customer membership list [39]. Morgan's Table [40] determined the minimum required respondents were 384 for this perception-based study (95% Confidence Interval (CI)). A total of 462 university students participated (completed the questionnaire) in the survey, where approximately 500 students were approached. Thus 92.40% was the overall participation rate for this study. However, we only considered 449 unvaccinated university students in the final analysis. To maintain the ethical issue of participant anonymity, we did not use cookies or the IP addresses of the participants' devices. Nevertheless, the study team monitored and double-checked the data on a frequent basis in order to avoid biases.

Data management and statistical analyses were performed by using Python (version 2.7; Beaverton, OR 97008, USA) and RStudio (version 1.2.5042; Boston, MA, USA) [41, 42]. Descriptive statistics such as the frequency and percentages were calculated where appropriate. Total knowledge and attitude scores were calculated by summing of respective item scores. A positive level in the knowledge and attitude section was determined based on the 80% cut-off scores [43]. For example, 7.2 was calculated as 80% score in the knowledge section. Thus, 7 and more than that was considered as positive knowledge level. Positive level/hesitancy and negative level/no hesitancy were scored 1 and 0, respectively. The Spearman's Rank correlation was carried out to determine the correlation in the KAH domain. Univariate and multiple logistic regression analyses were also performed to determine the predictors of knowledge, attitude, and hesitancy.

## Results

Table 1 summarizes the socio-demographic and academic information of the participants. This study had about an equal proportion of male (51.67%) and female (48.33%) participants. The majority of the students were unmarried (93.10%), living with their family (90.42%), living in Dhaka city (73.94%). 63.70% were from government-funded public universities. 21.60% of the participants tested COVID-19 positive before.

Table 2 demonstrates that university students used the internet (34.74%), social media (33.41%), and electronic media (25.61%) for COVID-19 vaccine-related information. The internet is a massive network that links devices (such as computers and smartphones) located worldwide. Individuals may exchange information and converse through the internet from any location with an internet connection. We defined social media as electronic modes of communication in which users build online communities to exchange information, ideas, personal messages, and other material. They are the foundations for the interactive web. They accomplish this by encouraging users to engage in, comment on, and create content. For example, Facebook. Electronic media is any form of media that can be shared on any electronic device for the purpose of viewing by an audience; unlike static media (printing), electronic media is transmitted to a larger audience. Electronic media include television (TV) and radio.

**Table 1. Socio-demographic and academic information (n = 449).**

| Features | n | % |
|---|---|---|
| 1. Gender | | |
| • Male | 232 | 51.67 |
| • Female | 217 | 48.33 |
| 2. Marital status | | |
| • Married | 31 | 6.90 |
| • Unmarried | 418 | 93.10 |
| 3. Living with family | | |
| • Yes | 406 | 90.42 |
| • No | 43 | 9.58 |
| 4. Current accommodation | | |
| • Dhaka city | 332 | 73.94 |
| • Outside Dhaka city | 117 | 26.06 |
| 5. University type | | |
| • Public | 286 | 63.70 |
| • Private | 163 | 36.30 |
| 6. University year | | |
| • 1st and 2nd year | 117 | 26.06 |
| • 3rd year | 200 | 44.54 |
| • 4th year and Masters | 132 | 29.40 |
| 7. Degree Major | | |
| • Arts and Social Science | 121 | 26.95 |
| • Business and Economics | 69 | 15.37 |
| • Science and Engineering | 141 | 31.40 |
| • Security and Strategic Studies | 57 | 12.69 |
| • Medical Studies | 61 | 13.59 |
| 8. COVID-19 positive before | | |
| • Yes | 97 | 21.60 |
| • No | 352 | 78.40 |

Table 3 presents the positive responses regarding the health behavior even after the vaccination (87.97%), a website for vaccine registration in Bangladesh (83.52%), side effects due to the vaccination (87.75%), and these side effects normally go in a few days (72.38%). 57.02% of respondents also reported that the COVID-19 vaccine is effective against the disease, and 68.82% believed that vaccination could help from severe illness due to the COVID-19. 69.71% of individuals thought that the full vaccination might help do everyday activities. 58.80% and 55.23% of the participants positively replied that infertility and long-term physical problems due to the COVID-19 vaccine are misconceptions.

**Table 2. Most used sources of COVID-19 related information.**

| Sources of COVID-19 related information | n (%) |
|---|---|
| Internet | 156 (34.74) |
| Social media | 150 (33.41) |
| Electronic media (TV, Radio) | 115 (25.61) |
| University | 18 (4.01) |
| People (community, family members) | 6 (1.30) |
| Print media | 4 (0.87) |

**Table 3. Knowledge towards COVID-19 vaccination (*n* = 449).**

| Statements | Positive Responses | |
|---|---|---|
| | *n* (%) | 95% CI[a] |
| COVID-19 vaccine is effective against the infection | 256 (57.02) | 53.4–61.6 |
| Health behaviour is required to follow even after vaccination | 395 (87.97) | 84.9–90.9 |
| Vaccinated can help from serious illness due to the COVID-19 | 309 (68.82) | 64.5–73.1 |
| Full vaccination may help to do normal activities | 313 (69.71) | 65.4–73.9 |
| Bangladeshi can register for COVID-19 vaccination through a website | 375 (83.52) | 80.0–86.9 |
| COVID-19 vaccine may have some side effects like other vaccines | 394 (87.75) | 84.7–90.7 |
| If there is any side effect due to the COVID-19 vaccination, they normally go in a few days | 325 (72.38) | 68.2–76.5 |
| COVID-19 vaccine can cause infertility | 264 (58.80) | 60.4–69.2 |
| COVID-19 vaccine can cause long-term physical problems | 248 (55.23) | 59.2–68.2 |

[a]CI = Confidence Intervals.

85.75% of participants were concerned about the pandemic, whereas 54.34% agreed on the safety and efficacy of the vaccine (Table 4). About 65% of individuals agreed that they and their family members should take the vaccine. However, less than 50% (43.88%) of the university students agreed that the COVID-19 vaccination could stop the ongoing pandemic.

Table 5 shows the association in the KAH domain. Knowledge and attitude had positive correlation (r = 0.452), whereas knowledge (r = -0.453) and attitude (r = -0.338) both were negatively correlated with the hesitancy. Increased odds of having a positive attitude (OR: 7.60; 95% CI: 4.94–11.85) were found for positive knowledge. Decreased odds of having hesitancy were recorded for the students with positive knowledge (OR: 0.10; 95% CI: 0.06–0.17) and a positive attitude (OR: 0.20; 95% CI: 0.13–0.32).

Overall, 58.13% and 64.81% of university students reported positive knowledge and attitude towards the COVID-19 vaccine. Table 6 summarizes the univariate and multiple logistic regression analysis results for knowledge and attitude level. University type and degree major were determined as significant predictors of knowledge level. Public university students reported decreased odds of having positive knowledge (OR: 0.19; 95% CI: 0.12–0.30) than private university students. Conversely, increased odds of having a positive knowledge were found among the students majoring in Business and Economics (OR: 2.39; 95% CI: 1.31–4.41), Science and Engineering (OR: 5.69; 95% CI: 3.36–8.81), and Medical studies (OR: 27.80; 95% CI: 10.52–96.45) compared to the students from Arts and Social Science. All the significant predictors in univariate analyses remained significant in multiple logistic regression analyses. Decreased adjusted odds ratio were determined in the case of public university students (aOR: 0.35; 95% CI: 0.19–0.63). Increased adjusted odds ratio was calculated for the Science

**Table 4. Attitude towards COVID-19 vaccination (*n* = 449).**

| Statements | Positive Attitudes | |
|---|---|---|
| | *n* (%) | 95% CI[a] |
| Concerned about the COVID-19 | 385 (85.75) | 82.5–88.9 |
| COVID-19 vaccine is safe and effective | 244 (54.34) | 49.7–58.9 |
| I need to take the vaccine as soon as possible if I am eligible | 295 (65.70) | 61.3–70.1 |
| I should aware and motivate my family members and neighbours to take the vaccine | 289 (64.37) | 59.9–68.8 |
| COVID-19 vaccination will help us to stop the pandemic | 197 (43.88) | 39.3–48.5 |

[a]CI = Confidence Intervals.

**Table 5. Association in knowledge, attitude, and hesitancy domain towards COVID-19 vaccine (*n* = 449).**

| Association | r-value[b] | OR[c] (95% CI) |
|---|---|---|
| Knowledge vs Attitude | 0.452*** | Positive Knowledge 7.60 (4.94–11.85) *** |
| Knowledge vs Hesitancy | -0.453*** | Positive Knowledge 0.10 (0.06–0.17) *** |
| Attitude vs Hesitancy | -0.338*** | Positive Attitude 0.20 (0.13–0.32) *** |

*p<0.05

**p<0.01

***p<0.001.

[b]r-value = correlation coefficient.

[c]OR = Odds Ratio.

**Table 6. Predictors of knowledge and attitude towards COVID-19 vaccine (*n* = 449).**

| Predictors | Knowledge | | Attitude | |
|---|---|---|---|---|
| | OR[c] (95% CI) | aOR[d] (95% CI) | OR[c] (95% CI) | aOR[d] (95% CI) |
| 1. Gender | | | | |
| • Female | 1 | | 1 | |
| • Male | 1.00 (0.69–1.46) | | 0.91 (0.62–1.34) | |
| 2. Marital status | | | | |
| • Married | 1 | | 1 | |
| • Unmarried | 0.45 (0.19–1.01) | | 1.36 (0.63–2.83) | |
| 3. Living with family | | | | |
| • No | 1 | | 1 | |
| • Yes | 1.67 (0.89–3.18) | | 1.10 (0.56–2.09) | |
| 4. Current accommodation | | | | |
| • Dhaka city | 1 | | 1 | |
| • Outside Dhaka city | 0.95 (0.62–1.46) | | 1.18 (0.76–1.85) | |
| 5. University type | | | | |
| • Private | 1 | | 1 | |
| • Public | 0.19 (0.12–0.30) *** | 0.35 (0.19–0.63) *** | 0.64 (0.42–0.96) * | |
| 6. University year | | | | |
| • 1st and 2nd year | 1 | | 1 | |
| • 3rd year | 1.57 (0.99–2.49) | | 1.07 (0.67–1.72) | |
| • 4th year and Masters | 1.46 (0.89–2.43) | | 1.29 (0.76–2.18) | |
| 7. Degree Major | | | | |
| • Arts and Social Science | 1 | | 1 | |
| • Business and Economics | 2.39 (1.31–4.41) ** | 1.28 (0.62–2.61) | 0.58 (0.32–1.07) | 0.53 (0.27–1.06) |
| • Science and Engineering | 5.69 (3.36–9.81) *** | 4.26 (2.46–7.49) *** | 1.01 (0.60–1.71) | 0.97 (0.56–1.68) |
| • Security and Strategic Studies | 1.05 (0.54–2.03) | 1.13 (0.57–2.18) | 0.32 (0.16–0.61) *** | 0.32 (0.17–0.62) *** |
| • Medical Studies | 27.80 (1052–96.45) *** | 12.79 (4.40–47.08) *** | 2.75 (1.27–6.47) * | 2.45 (1.01–6.35) |
| 8. COVID-19 positive before | | | | |
| • No | 1 | | 1 | |
| • Yes | 1.29(0.81–2.06) | | 0.95(0.60–1.53) | |

*p<0.05

**p<0.01

***p<0.001.

[c]OR = Odds Ratio.

[d]aOR = Adjusted Odds Ratio.

**Table 7. Predictors of hesitancy towards COVID-19 vaccine ($n$ = 449).**

| Predictors | Hesitancy | |
|---|---|---|
| | OR[c] (95% CI) | aOR[d] (95% CI) |
| 1. Gender | | |
| • Female | 1 | |
| • Male | 1.02 (0.67–1.56) | |
| 2. Marital status | | |
| • Married | 1 | |
| • Unmarried | 2.50 (0.95–8.60) | |
| 3. Living with family | | |
| • No | 1 | |
| • Yes | 1.18 (0.58–2.60) | |
| 4. Current accommodation | | |
| • Dhaka city | 1 | |
| • Outside Dhaka city | 0.81 (0.49–1.31) | |
| 5. University type | | |
| • Private | 1 | |
| • Public | 4.62 (2.70–8.31) *** | 2.29 (1.19–4.55) * |
| 6. University year | | |
| • 1st and 2nd year | 1 | |
| • 3rd year | 0.53 (0.32–0.88) * | |
| • 4th year and Masters | 0.67 (0.38–1.15) | |
| 7. Degree Major | | |
| • Arts and Social Science | 1 | |
| • Business and Economics | 0.18 (0.08–0.38) *** | 0.30 (0.12–0.67) ** |
| • Science and Engineering | 0.29 (0.17–0.50) *** | 0.37 (0.20–0.68) ** |
| • Security and Strategic Studies | 0.42 (0.21–0.82) * | 0.43 (0.21–0.84) * |
| • Medical Studies | 0.06 (0.01–0.16) *** | 0.10 (0.02–0.35) *** |
| 8. COVID-19 positive before | | |
| • No | 1 | |
| • Yes | 0.58 (0.32–1.00) | |

*$p < 0.05$

**$p < 0.01$

***$p < 0.001$.

[c]OR = Odds Ratio.

[d]aOR = Adjusted Odds Ratio.

and Engineering (aOR: 4.26; 95% CI: 2.46–7.49) and Medical Studies (aOR: 12.79; 95% CI: 4.40–47.08) students. Univariate analyses also determined the university type and degree major as significant predictors of attitude level. Public university students (OR: 0.64; 95% CI: 0.42–0.96) reported decreased odds of having a positive attitude. Students from Security and Strategic Studies reported decreased odds of having a positive attitude (OR: 0.32; 95% CI: 0.16–0.61). In contrast, Medical Studies students demonstrated increased odds of having a positive attitude (OR: 2.75; 95% CI: 1.27–6.47). Multiple analyses determined only the degree major as a significant predictor of attitude level. Decreased adjusted odds of having a positive attitude were found for the Security and Strategic Studies students (aOR: 0.32; 95% CI: 0.17–0.62).

**Table 8. Reasons of hesitancy towards COVID-19 vaccine ($n$ = 117).**

| Statements | Hesitated | |
|---|---|---|
| | *n* (%)) | 95% CI |
| Fear of ineligibility | 46 (39.32) | 30.3–48.3 |
| Lack of information | 83 (70.94) | 62.6–79.3 |
| Fear of side effects | 102 (87.18) | 81.0–93.3 |
| Probable cost of vaccination | 07 (5.98) | 1.6–10.3 |
| Against religious beliefs | 03 (2.56) | -0.3–5.4 |
| Other reasons | 53 (45.30) | 36.1–54.4 |

[a]CI = Confidence Intervals.

Univariate analyses determined the university type, university year, and degree major as significant predictors of hesitancy (Table 7). Increased odds of having hesitancy were found in public university students (OR: 4.62; 95% CI: 2.70–8.31) compared to the private university students. Decreased odds of having hesitancy were calculated when compared the 3$^{rd}$ year students (OR: 0.53; 95% CI: 0.32–0.88) with 1$^{st}$ and 2$^{nd}$ year students; compared the students major in Business and Economics (OR: 0.18; 95% CI: 0.08–0.38), Science and Engineering (OR: 0.29; 95% CI: 0.17–0.50), Security and Strategic Studies (OR: 0.42; 95% CI: 0.21–0.82) and Medical Studies (OR: 0.06; 95% CI: 0.01–0.16) with Arts and Social Science major students. All the significant predictors, except the university year, determined in univariate analyses remained significant in multiple analyses. Increased adjusted odds ratios were recorded for public university students (aOR: 2.29; 95% CI: 1.19–4.55). Conversely, decreased adjusted odds ratio were found in case of the students from Business and Economics (aOR: 0.30; 95% CI: 0.12–0.67), Science and Engineering (aOR: 0.37; 95% CI: 0.20–0.68), Security and Strategic Studies (aOR: 0.43; 95% CI: 0.21–0.84) and Medical Studies (aOR: 0.10; 95% CI: 0.02–0.35).

Overall, 26.06% ($n$ = 117) of the participants showed their hesitancy towards the COVID-19 vaccine. Table 8 shows that the fear of side effects (87.18) and lack of information (70.94%) were recorded as major reasons for the hesitancy.

## Discussion

University students in Bangladesh have already been greatly affected due to the COVID-19 [28, 44, 45]. It is vital to include university students in the COVID-19 vaccination program. However, no research has been done on their responses to the new COVID-19 vaccination. This study, we believe, is the first to assess their KAH regarding the vaccine.

This study indicates that students understand the need to maintain healthy habits (mask, handwashing, and avoiding crowds) even after the 1$^{st}$ dose of vaccine. However, many of these students did not believe the available COVID-19 vaccines, and most were worried about the pandemic. Other research revealed the pandemic's impact on academics and mental health [28].

We found that positive knowledge and attitude may reflect vaccination hesitancy among university students. But many of these students had negative knowledge and attitudes. The authority should boost vaccination knowledge before the inoculation campaign, especially for university students [30]. This study indicated that many students were unaware of vaccination myths. Misconceptions about the safety and efficacy of the vaccine could also delay inoculation campaigns [12]. Media has always been an essential factor in enhancing perception, which was also found during COVID-19 [46]. Our study identified that the internet, social media, and electronic media (TV, Radio) played a major role in reaching university students. It also

supports another study [47] which found that most public university students had COVID-19 relevant information from electronic media followed by social media and the internet. These modern media can play a vital role in disseminating authentic COVID-19 vaccine information. Web-based and mobile applications can also be positive media for this purpose. The study already indicated the increasing popularity of the internet and social media in developing countries [48]. Social media has already contributed [49] substantially during the pandemic where social distancing was the primary concern.

Socio-demographic and academic information is essential to track university students. Thus, evaluating the association between this information and the KAH domain level becomes crucial. Our study found that only academic information is the predictor of knowledge, attitude, and hesitancy towards the COVID-19 vaccine. Several studies on COVID-19 vaccine hesitancy among Bangladeshi respondents evaluated socio-demographic information, behavioral predictors, and academic information [18–20]. One study conducted among Czech university students found socio-demographic information such as gender and nationality as significant predictors of COVID-19 vaccine hesitancy [16]. They also found that knowledge level and dependent media as significant predictors of vaccine hesitancy. Another study conducted among Italian university students also found gender, major, and lower academic level as significant predictors of vaccine hesitancy [50]. University rank was also found as significant predictor of vaccine hesitancy among university students of Lebanon [17]. Our study found that the public university students reported negative knowledge and attitude compared to private university students. They also showed hesitancy towards the vaccine. In Bangladesh, public university students are among the most affected student groups during the pandemic [47]. They have been suffering due to the long-term closure period of the university. A large number of economically deprived students with poor internet networks placed many public universities in challenging situations. They faced challenges to start online academic activities where most private universities have already started it [28, 51]. Even with several challenges [52], online academic activities might improve the perception level among private university students. Bangladesh's government has already taken initiatives to ensure online educational activities in all universities [53]. Medical Studies students showed positive knowledge, positive attitude, and thereby low hesitancy. These students might have better access to authentic COVID-19 vaccine-related information compared to other majoring students. They play a critical role in the vaccination campaign by disseminating reliable information regarding the effectiveness and safety of COVID-19 vaccines [54, 55]. Their health-related knowledge and attitudes are at the pinnacle of the student body, making them the go-to people for advice on public health concerns [55, 56]. However, they may have drivers of vaccine hesitancy, such as side effects of new vaccines and available vaccines in the local health center [55].

This study revealed some responsible factors for the hesitancy regarding the COVID-19 vaccine, which can be addressed as followings:

- Fear of side effects and a lack of information were identified as the top two reasons for hesitancy about the COVID-19 vaccine. The government and appropriate authorities should examine various measures to address these issues. A holistic strategy should be taken, in which individuals from all sectors, including university administration, collaborate to vaccinate a significant number of university students.

- Collaboration between researchers, universities, the government, and other relevant entities should be taken into account. Researchers might give information on the vaccination status, challenges, and strategic solutions, assisting the government and university authorities in implementing a successful vaccination program. Numerous studies have been done to

establish the COVID-19 vaccine's safety on an independent basis, with the objective of lowering public vaccination hesitancy [57, 58].

- Responsible authorities may consider using a variety of communication platforms, including the internet, social media, and electronic media, to distribute accurate information about the COVID-19 vaccine. They may also utilize mobile apps to reach university students who are avid users of various mobile applications.

- The university administration should treat pandemic as a serious concern. Their students should be knowledgeable about it. They are capable of incorporating pandemic-related studies into any type of major. They have the option of revising the present curriculum. Additionally, the government can aid in this endeavor.

- Finally, our findings underscore the critical nature of the COVID-19 awareness campaign in order to ensure a timely and successful vaccination program. Sufficient accurate information regarding the vaccination can boost confidence.

Like other studies, this study has some limitations. Due to the ongoing pandemic, we considered a self-reported online survey that might not reach respondents without internet access. This study required a rapid survey to examine the responses during the mass administration of the COVID-19 vaccine in Bangladesh. It was also conducted before the inoculation campaign for university students. The perception level and hesitancy might change over time, where Bangladesh authorities have already been considering several available COVID-19 vaccines [59]. However, this exploratory study provides essential information for making the inoculation campaign effective. Further research should be conducted to evaluate the perception and hesitancy level over time.

## Conclusion

This study provides useful information for the successful inoculation campaign among university students. The success of the inoculation campaign largely depends on the hesitancy towards the vaccine. The low hesitancy can be possible by disseminating enough authentic information among these students. Since the university students mostly used the internet, social media, and electronic media for COVID-19 vaccine-related information, extensive campaigns through these media should be incorporated along with the inoculation campaign. Collaborations between the university and authority are also required. Furthermore, the university should revise the existing curriculum to ensure the pandemic relevant subject and activities. It can also prepare the whole community for the future pandemic since university students can act as a hub to reach their surrounding communities.

## Supporting information

**S1 Questionnaire.**
(DOCX)

## Acknowledgments

We highly appreciate the university students who participated in this study. We would also like to mention the university students and experts who support the research.

## Author Contributions

**Conceptualization:** Md. Mostafizur Rahman, Musabber Ali Chisty, Md. Ashraful Alam, Mohammed Sadman Sakib, Masrur Abdul Quader, Ifta Alam Shobuj, Md. Abdul Halim, Farzana Rahman.

**Data curation:** Md. Mostafizur Rahman, Mohammed Sadman Sakib, Masrur Abdul Quader, Ifta Alam Shobuj, Md. Abdul Halim, Farzana Rahman.

**Formal analysis:** Md. Mostafizur Rahman.

**Investigation:** Md. Mostafizur Rahman.

**Methodology:** Md. Mostafizur Rahman, Musabber Ali Chisty, Md. Ashraful Alam.

**Project administration:** Md. Mostafizur Rahman.

**Resources:** Md. Mostafizur Rahman, Musabber Ali Chisty, Md. Ashraful Alam.

**Software:** Md. Mostafizur Rahman.

**Supervision:** Md. Mostafizur Rahman.

**Validation:** Md. Mostafizur Rahman.

**Visualization:** Md. Mostafizur Rahman.

**Writing – original draft:** Md. Mostafizur Rahman.

**Writing – review & editing:** Md. Mostafizur Rahman, Musabber Ali Chisty, Md. Ashraful Alam, Mohammed Sadman Sakib, Masrur Abdul Quader, Ifta Alam Shobuj, Md. Abdul Halim, Farzana Rahman.

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
