## [Decision Letter · Decision Letter 0]

6 Dec 2021

PONE-D-21-17512Knowledge, attitude, and hesitation towards covid-19 vaccine among university students of BangladeshPLOS ONE

Dear Dr. Rahman,

Thank you for submitting your manuscript to PLOS ONE. After careful consideration, we feel that it has merit but does not fully meet PLOS ONE’s publication criteria as it currently stands. Therefore, we invite you to submit a revised version of the manuscript that addresses the points raised during the review process.

 Three reviewers have evaluated your submission and have identified a range of points that need to be addressed in relation to the description of the study designs and the presentation of your manuscript. Please respond carefully to all of the reviewers concerns when preparing your revisions.

We look forward to receiving your revised manuscript.

Kind regards,

Jamie Males

Staff Editor

PLOS ONE

Journal Requirements:

3. Thank you for stating the following financial disclosure: "They was no external fund for this research." 

Reviewers' comments:

Reviewer's Responses to Questions

**Comments to the Author**

1. Is the manuscript technically sound, and do the data support the conclusions?

Reviewer #1: Yes

Reviewer #2: Partly

Reviewer #3: Yes

2. Has the statistical analysis been performed appropriately and rigorously? 

Reviewer #1: No

Reviewer #2: Yes

Reviewer #3: Yes

3. Have the authors made all data underlying the findings in their manuscript fully available?

Reviewer #1: No

Reviewer #2: Yes

Reviewer #3: No

4. Is the manuscript presented in an intelligible fashion and written in standard English?

Reviewer #1: Yes

Reviewer #2: Yes

Reviewer #3: Yes

5. Review Comments to the Author

Reviewer #1: Dear author(s),

Thank you for your esteemed efforts that aimed to increase our collective knowledge about the ongoing situation COVID-19 vaccine hesitancy among young adults population.

Please consider the following minor comments:

1. Add operational (working definitions) for electronic media, social media resources, etc.

2. Use CHERRIS guidelines in structring and reporting your study and cite it your Methods section.

Ref: https://pubmed.ncbi.nlm.nih.gov/15471760

4. It is not clear nor validated how the question about "relevant subject to vaccine" can be used in this situation. Of course, the level of knowledge that a medical students in terms of infectious diseases and immunization is not comprable to the level of knowledge that a student in humantrian sciences will get. I would suggest you to omit this question from your manuscript and your analysis.

5. In Table 2, please correct (06) and (04). Remove the zero.

6. In Table 3, please add a dash in the coloumn of CI instead of the word "to".

7. In Line 169, you mentioned for the first time in the manuscript that a small fraction of your sample was already vaccinated. As long this small fraction is not useful for downstream analysis, I would strongly recommend you to remove them from the begining and exclude them because their inclusion will overestimate the results.

8. Line 271 - 281: you can compare your results to other results from different countries university students.

Ref: https://www.ncbi.nlm.nih.gov/pmc/articles/PMC8470400

9. Line 282: you can mention the role of medical students as well within the context of public campaigns for vaccines and their particular drivers of vaccines hesitancy during COVID-19.

Ref: https://pubmed.ncbi.nlm.nih.gov/34072500

10. Line 292, you can mention the COVID-19 vaccine safety studies that aim to provide independent evidence on the COVID-19 vaccine safety that can suppress the public levels of vaccine hesitancy.

Ref: https://www.ncbi.nlm.nih.gov/pmc/articles/PMC8345554

Ref: https://pubmed.ncbi.nlm.nih.gov/34577573

Sincerely,

Reviewer #2: This manuscript is noteworthy in the current era of COVID-19 pandemic because the issue of vaccine hesitancy is extremely important as WHO considers vaccine hesitancy as a significant threat to global health. The COVID-19 vaccine is one of the top priorities in the battle against the pandemic, but its effectiveness is dependent on individual decisions since vaccines are voluntary. Apart from scarcity in dose availability and inequity in vaccine delivery, vaccine hesitancy is one of the most serious challenges in this regard. A main finding of the study, that there is still a considerable vaccine hesitancy among university students despite their goog knowledge and positive attitude, which is consistent with studies in other countries and across broad demographic groups.

However, some issues needed to be addressed first to improve the quality of the manuscript: though, the following are general outlines of the required modifications:

Title: It is better to replace "hesitation" with "hesitancy" also along the whole manuscript.

- Language editing is needed for multiple issues

- Introduction:

- It would be good to provide a background on COVID-specific reasons for vaccine hesitancy, along with the general ones.

- Some background on vaccination plans in the Bangladesh should be displayed such as What vaccines are being considered, is the vaccination program already planned, number of currently vaccinated person. What is the perception of vaccines in Bangladesh, e.g., in the news? These are all crucial factors playing a role in vaccine hesitancy and affecting its dynamics.

- The rationale, study hypothesis and the study questions should be clearly stated

Methodology section: issues in the study design and settings.

Methods:

- " The current survey was performed among university students"

- Each group should be defined; by mentioning the study plan of each faculty and whether their curriculum contain COVI-19 vaccination topics or not?

- Also sample size calculation and sampling technique (details of methods of selection of participants) are not mentioned in the manuscript.

-

- Results:

- What is the age of the participants?

Knowledge leveling is either Good or poor or adequate & inadequate NOT positive or negative as the attitude

What is the level of COVID-19 vaccination hesitancy? Where this percentage in the results?? It needs a separate table as knowledge and attitude as it is one if the study objectives

- Discussion:

- Generally, the discussion is somewhat redundant without adequate structure and should be focused to compare the results with similar studies and to explain and give the implications of the study results.

- Please make the discussion more concise, without discussing too many details but the most important results and their general meaning.

- Having some bullet points on how to address the observed hesitancy would be beneficial for the discussion and the entire manuscript.

- References:

- Needs revision as sometimes they are not consistent in their style. Some of them are incomplete

Reviewer #3: This is a well-written manuscript that evaluates hesitancy towards the COVID-19 vaccine in University students in Bangladesh. The results are consistent with several other studies in other locations, if not entirely novel.

My minor concern is that the university student population may be very different from the population at large. Although this is clearly a manuscript about university students, some effort to compare the university population to the population of Bangladesh as a whole would be appreciated.

6. PLOS authors have the option to publish the peer review history of their article (what does this mean?). If published, this will include your full peer review and any attached files.

Reviewer #1: No

Reviewer #2: No

Reviewer #3: No

---

## [Author Response · Author response to Decision Letter 0]

25 Feb 2022

Reviewer #1:

Dear author(s), Thank you for your esteemed efforts that aimed to increase our collective knowledge about the ongoing situation COVID-19 vaccine hesitancy among young adults population.

Please consider the following minor comments:

1. Add operational (working definitions) for electronic media, social media resources, etc.

Response: Thank you for the comment. We have added operational definitions for the internet, electronic media, and social media after Table 2 in Results section.

2. Use CHERRIS guidelines in structring and reporting your study and cite it your Methods section.

Ref: https://pubmed.ncbi.nlm.nih.gov/15471760

Response: Thank you for the suggestion. We have revised our Methods following CHERRIS guidelines. We have also cited it in our Methods section.

4. It is not clear nor validated how the question about "relevant subject to vaccine" can be used in this situation. Of course, the level of knowledge that a medical students in terms of infectious diseases and immunization is not comprable to the level of knowledge that a student in humantrian sciences will get. I would suggest you to omit this question from your manuscript and your analysis.

Response: Thank you for pointing out it. We have omitted that question from our manuscript and our analysis. 

5. In Table 2, please correct (06) and (04). Remove the zero.

Response: Thank you. We have corrected those numbers.

6. In Table 3, please add a dash in the coloumn of CI instead of the word "to".

Response: Thank you. We have replaced the word “to” by dash in the column of CI for all Table (including Table 3) to maintain the consistency.

7. In Line 169, you mentioned for the first time in the manuscript that a small fraction of your sample was already vaccinated. As long this small fraction is not useful for downstream analysis, I would strongly recommend you to remove them from the begining and exclude them because their inclusion will overestimate the results.

Response: Thank you for the recommendation. We have removed that small fraction from our sample who were vaccinated. We have mentioned it in our Methods, and revised our Results from the beginning.

8. Line 271 - 281: you can compare your results to other results from different countries university students.

Ref: https://www.ncbi.nlm.nih.gov/pmc/articles/PMC8470400

Response: Thank you. We have revised that part following your comment. We have also added the above reference.

9. Line 282: you can mention the role of medical students as well within the context of public campaigns for vaccines and their particular drivers of vaccines hesitancy during COVID-19.

Ref: https://pubmed.ncbi.nlm.nih.gov/34072500

Response: Thank you. We have revised and cited some new references following the abovementioned reference.

10. Line 292, you can mention the COVID-19 vaccine safety studies that aim to provide independent evidence on the COVID-19 vaccine safety that can suppress the public levels of vaccine hesitancy.

Ref: https://www.ncbi.nlm.nih.gov/pmc/articles/PMC8345554

Ref: https://pubmed.ncbi.nlm.nih.gov/34577573

Response: Thank you very much for all effective comments and suggestions which have improved the quality of our manuscript a lot. We have revised with the abovementioned references following you and another reviewer’s comments. 

Reviewer #2: 

This manuscript is noteworthy in the current era of COVID-19 pandemic because the issue of vaccine hesitancy is extremely important as WHO considers vaccine hesitancy as a significant threat to global health. The COVID-19 vaccine is one of the top priorities in the battle against the pandemic, but its effectiveness is dependent on individual decisions since vaccines are voluntary. Apart from scarcity in dose availability and inequity in vaccine delivery, vaccine hesitancy is one of the most serious challenges in this regard. A main finding of the study, that there is still a considerable vaccine hesitancy among university students despite their goog knowledge and positive attitude, which is consistent with studies in other countries and across broad demographic groups.

However, some issues needed to be addressed first to improve the quality of the manuscript: though, the following are general outlines of the required modifications:

Title: It is better to replace "hesitation" with "hesitancy" also along the whole manuscript.

Response: Thank you very much for appreciating our work which will encourage us to conduct further research. We have replaced "hesitation" with "hesitancy" in both title and whole manuscript.

- Language editing is needed for multiple issues

Response: Thank you. We have done major revision and checked the language. We hope the current version would satisfy you.

-Introduction:

- It would be good to provide a background on COVID-specific reasons for vaccine hesitancy, along with the general ones.

Response: Thank you again. We have revised and provide background on COVID-specific reasons for vaccine hesitancy, along with general ones.

- Some background on vaccination plans in the Bangladesh should be displayed such as What vaccines are being considered, is the vaccination program already planned, number of currently vaccinated person. What is the perception of vaccines in Bangladesh, e.g., in the news? These are all crucial factors playing a role in vaccine hesitancy and affecting its dynamics.

Response: Thank you. We have revised and added the information following your comment. Please check the introduction part.

- The rationale, study hypothesis and the study questions should be clearly stated

Response: Thank you. We have revised again following your comment. We hope that the current revised version will satisfy you.

Methodology section: issues in the study design and settings.

Methods:

- " The current survey was performed among university students"

- Each group should be defined; by mentioning the study plan of each faculty and whether their curriculum contain COVI-19 vaccination topics or not?

Response: Thank you for pointing out it. Actually, we have divided 5 groups based on the major considering Bangladesh perspective. We wanted to examine if there were any differences among these groups. All these groups have diverse study plan and curriculum. For instance, Science and Engineering includes several study plans based on the university authority. However, we have revised our Research design following your comment. We discussed the differences among these groups based on our hypothesis in Methods and Discussion section. We have also deleted “COVID-19 relevant subject” from our manuscript following another reviewer’s suggestion. 

- Also sample size calculation and sampling technique (details of methods of selection of participants) are not mentioned in the manuscript.

Response: Thank you. We have revised and put the information regarding sampling technique in Data collection and data analysis section as “We followed the convenience sampling technique. Convenience sampling is a technique for non-probabilistic sampling that is frequently used in clinical research. Typically, this sampling strategy draws clinical cases or participants from an easily accessible area, such as a medical records database, an internet site, or a customer membership list [39].” For sample size, we followed Morgan’s Table which determined the minimum required respondents were 384 for this perception-based study (95% Confidence Interval (CI)). We have also mentioned it in Data collection and data analysis section.

-Results:

- What is the age of the participants?

Response: Thank you. The age of this young adult was 18-25 years. Since, majority of them were in same age group, we did not consider in our analysis.

Knowledge leveling is either Good or poor or adequate & inadequate NOT positive or negative as the attitude

What is the level of COVID-19 vaccination hesitancy? Where this percentage in the results?? It needs a separate table as knowledge and attitude as it is one if the study objectives.

Response: Thank you again. We have mentioned that overall, 26.06% of the participants showed their positive hesitancy towards the COVID-19 vaccine. Hesitancy was measured based on their positive and negative response, and then the factors responsible for this hesitancy. Table 7 and Table 8 summarizes the result regarding hesitancy towards the COVID-19 vaccine. Table 8 also shows the percentages. 

-Discussion:

- Generally, the discussion is somewhat redundant without adequate structure and should be focused to compare the results with similar studies and to explain and give the implications of the study results.

Response: Thank you. We have revised and added some other similar studies following you and other reviewers’ comments. 

- Please make the discussion more concise, without discussing too many details but the most important results and their general meaning.

Response: Thank you. We have revised the discussion part following your comment. We may still have some discussion to clarify the understanding following other relevant studies.

- Having some bullet points on how to address the observed hesitancy would be beneficial for the discussion and the entire manuscript.

Response: Thank you. We have revised and provided some bullet points before limitation of the research. 

-References:

- Needs revision as sometimes they are not consistent in their style. Some of them are incomplete

Response: Thank you. We have revised again. We actually used referencing management tool. 

Reviewer #3: This is a well-written manuscript that evaluates hesitancy towards the COVID-19 vaccine in University students in Bangladesh. The results are consistent with several other studies in other locations, if not entirely novel.

My minor concern is that the university student population may be very different from the population at large. Although this is clearly a manuscript about university students, some effort to compare the university population to the population of Bangladesh as a whole would be appreciated.

Response: Thank you very much. We have done major revision following all reviewers’ comments and suggestions which also included COVID-19 vaccine information regarding general population of Bangladesh (please check both introduction and discussion part). We also have added some references which considered general population of Bangladesh.

---

## [Decision Letter · Decision Letter 1]

16 Jun 2022

Knowledge, attitude, and hesitancy towards COVID-19 vaccine among university students of Bangladesh

PONE-D-21-17512R1

Dear Dr. Rahman,

We’re pleased to inform you that your manuscript has been judged scientifically suitable for publication and will be formally accepted for publication once it meets all outstanding technical requirements.

Kind regards,

Jianhong Zhou

Staff Editor

PLOS ONE

Additional Editor Comments (optional):

Reviewers' comments:

Reviewer's Responses to Questions

**Comments to the Author**

1. If the authors have adequately addressed your comments raised in a previous round of review and you feel that this manuscript is now acceptable for publication, you may indicate that here to bypass the “Comments to the Author” section, enter your conflict of interest statement in the “Confidential to Editor” section, and submit your "Accept" recommendation.

Reviewer #1: All comments have been addressed

Reviewer #2: All comments have been addressed

2. Is the manuscript technically sound, and do the data support the conclusions?

Reviewer #1: Yes

Reviewer #2: Yes

3. Has the statistical analysis been performed appropriately and rigorously? 

Reviewer #1: Yes

Reviewer #2: Yes

4. Have the authors made all data underlying the findings in their manuscript fully available?

Reviewer #1: Yes

Reviewer #2: Yes

5. Is the manuscript presented in an intelligible fashion and written in standard English?

Reviewer #1: Yes

Reviewer #2: Yes

6. Review Comments to the Author

Reviewer #1: Dear Author(s),

Thank you for your esteemed efforts in responding to all my previous points. I do believe that the manuscript is in good shape now for being accepted. Good luck!

Sincerely,

Reviewer #2: Thanks for doing the modifications... The manuscript now is suitable to be published as most of the corrections were done as needed

7. PLOS authors have the option to publish the peer review history of their article (what does this mean?). If published, this will include your full peer review and any attached files.

Reviewer #1: No

Reviewer #2: No

---

## [Editor Report · Acceptance letter]

17 Jun 2022

PONE-D-21-17512R1 

Knowledge, attitude, and hesitancy towards COVID-19 vaccine among university students of Bangladesh 

Dear Dr. Rahman:

I'm pleased to inform you that your manuscript has been deemed suitable for publication in PLOS ONE. Congratulations! Your manuscript is now with our production department. 

Kind regards, 

on behalf of

Jianhong Zhou 

Staff Editor

PLOS ONE